# Fused Cells between Human-Adipose-Derived Mesenchymal Stem Cells and Monocytes Keep Stemness Properties and Acquire High Mobility

**DOI:** 10.3390/ijms23179672

**Published:** 2022-08-26

**Authors:** Karla Montalbán-Hernández, Cesar Casado-Sánchez, José Avendaño-Ortiz, José Carlos Casalvilla-Dueñas, Gloria C. Bonel-Pérez, Julia Prado-Montero, Jaime Valentín-Quiroga, Roberto Lozano-Rodríguez, Verónica Terrón-Arcos, Fátima Ruiz de la Bastida, Laura Córdoba, Fernando Laso-García, Luke Diekhorst, Carlos del Fresno, Eduardo López-Collazo

**Affiliations:** 1The Innate Immune Response Group, IdiPAZ, La Paz University Hospital, 28046 Madrid, Spain; 2Tumour Immunology Lab, IdiPAZ, La Paz University Hospital, 28046 Madrid, Spain; 3Department of Plastic and Reconstructive Surgery, University Hospital La Paz, 28046 Madrid, Spain; 4Centre for Biomedical Research Network of Respiratory Diseases (CIBERES), 28029 Madrid, Spain; 5Biobank Platform, IdiPAZ, La Paz Universitary Hospital, 28046 Madrid, Spain; 6Neurological Sciences and Cerebrovascular Research Laboratory, Department of Neurology and Stroke Center, Neurology and Cerebrovascular Disease Group, Neuroscience Area La Paz Hospital Institute for Health Research–IdiPAZ, 28046 Madrid, Spain

**Keywords:** mesenchymal stem cell, monocyte, fusion, migration, stemness

## Abstract

Human-adipose-derived mesenchymal stem cells (hADMSCs) are multipotent stem cells which have become of great interest in stem-cell therapy due to their less invasive isolation. However, they have limited migration and short lifespans. Therefore, understanding the mechanisms by which these cells could migrate is of critical importance for regenerative medicine. **Methods:** Looking for novel alternatives, herein, hADMSCs were isolated from adipose tissue and co-cultured with human monocytes ex vivo. **Results:** A new fused hybrid entity, a foam hybrid cell (FHC), which was CD90^+^CD14^+^, resulted from this co-culture and was observed to have enhanced motility, proliferation, immunomodulation properties, and maintained stemness features. **Conclusions:** Our study demonstrates the generation of a new hybrid cellular population that could provide migration advantages to MSCs, while at the same time maintaining stemness properties.

## 1. Introduction

Human-adipose-derived mesenchymal stem cells (hADMSCs) are multipotent stem cells which possess the abilities of self-renewal and differentiation to various cell types [1]. They can aid in tissue regeneration through their paracrine signalling, which involves the secretion of cytokines, chemokines, and growth factors [2]. Due to their safer and less invasive isolation compared to those isolated from the bone marrow and their differentiation capability towards chondrocytes, adipocytes, osteocytes, cardiomyocytes, and vascular endothelial cells, hADMSCs have become a breakthrough in regenerative medicine [3,4]. There is an increasing number of studies that have reported their use in graft versus host disease treatment, suggesting a potent immunomodulatory function [5].

The use of these cells in degenerative and inflammatory diseases has been based on their potential capability to migrate towards the site of injury and interact with inflammatory cells to mediate the resolution of inflammation, which is why their homing ability and engraftment potential are crucial for a correct regenerative response [6]. However, experimental evidence has shown the rapid disappearance of these cells after in vivo administration, raising questions as to their therapeutic potential [7]. According to several studies, this occurs because the MSCs are unable to migrate to the site of injury or have a short lifespan after administration [8,9]. Alternatively, Eggenhofer et al. (2014) postulate that a small fraction of these MSCs could be escaping death and having their beneficial effects at the injury site, or that they could also transfer their beneficial effects to other cells, which would subsequently mediate tissue repair or immunomodulation.

Previously, we and others postulated that mesenchymal stem cells have the ability to fuse with other cellular types, such as monocytes and macrophages, allowing the exchange of genetic information and, subsequently, some specific capabilities [10,11]. Cell–cell fusion is an important mechanism which regulates both physiological and pathological processes such as muscle development, fertilisation, tumorigenesis, and stem-cell-mediated tissue regeneration [12,13,14].

Interactions between M1-polarised monocytes and MSCs have been shown to enhance the immunosuppressive function of the MSCs [15], and monocytes/macrophages were observed to develop regulatory functions after phagocytosis of dead MSCs [16]. Numerous studies have demonstrated that MSCs influence macrophage function in the context of tissue repair and inflammation resolution after their mutual interaction [17]. Moreover, monocytes/macrophages are known to migrate towards tumour sites and fuse with cancer stem cells to produce new hybrid entities with migratory abilities and, therefore, enhanced metastatic potential [18]. These data point towards a complex interplay between MSCs and macrophages that needs further exploration.

Herein, we describe cell fusion between hADMSCs and monocytes ex vivo. We characterised these new hybrid cellular entities defined as Foam Hybrid Cells (FHCs) and studied their main characteristics. This new hybrid cell population could provide a novel explanation for the lack of MSC detection after administration. This fusion could be easily ignored and could simply provide a mechanism by which these MSCs acquire enhanced migratory abilities after infusion. Altogether, FHCs could become of great relevance in regenerative medicine with further experimentation, if proven to be as beneficial as we have anticipated.

## 2. Results

### 2.1. Co-Culture of hADMSCs with Human Monocytes Ex Vivo Yields a New Hybrid Entity

To test the potential fusion between hADMSCs and human monocytes, mesenchymal stem cells, isolated from human adipose tissue, and human monocytes were stained with the vital colorants DIO and DID, respectively. Then, these cells were co-cultured for 5 days, following previously established experimental conditions [18] (Figure 1A). Confocal microscopy confirmed the generation of double-positive entities that we defined as foam hybrid cells (FHCs, DID^+^DIO^+^) (Figure 1B,C). Note that, although the resulting fused cells exhibited a similar shape to the original hADMSCs, the diameter size was significantly larger in the FHCs compared to their parental cells (Figure 1C right panels, and D). In addition, after a 5-day co-culture between CD90^+^ hADMSCs and CD14^+^ human monocytes, flow cytometry analysis revealed a double-positive population (CD90^+^CD14^+^, Figure 1E). The gating strategy followed is shown in Appendix A.

### 2.2. Pro-Inflammatory Phenotype Favours FHC Formation

Once FHC (CD90^+^CD14^+^) generation as a result of hADMSCs co-culture with human monocytes was established, we moved to study the influence of the monocyte-inflammatory status on this phenomenon. Then, purified human monocytes were polarised towards M1 and M2 phenotypes using IFNγ and IL4, respectively (Figure 2A). Polarisation was verified by both cell-surface-markers’ expression and cytokine production. On one hand, the HLA-DR expression was confirmed on M1 monocytes and CD163 on M2 monocytes (Figure 2B,C), and, on the other hand, supernatants from M1 monocytes showed significantly higher levels of IL6, IL8, and TNFα (Figure 2D–F), in contrast, M2 monocytes produced significantly higher levels of IL4, IL10, and TGFβ (Figure 2G–I). Supernatants were collected at 48 h, after a 24 h stimulation followed by a wash and 24 h resting. Note that, despite the high CD163 and HL-DR mean fluorescence intensity (MFI) on M0 monocytes, no secretion of IL6, TNFα, IL4, IL10, or TGFβ, respectively, were detected in supernatants of these cell cultures. Finally, the co-culture of hADMSCs with M1 polarised monocytes showed a higher percentage of FHCs when compared to M2 and M0 (Figure 2J,K).

### 2.3. Foam Hybrid Cells Generated after Co-Culture Exhibit Strong Immunomodulatory Features

To study the potential immunomodulatory properties of FHCs, we evaluated the expression of the immune-checkpoint (IC) ligands SIGLEC5 and PD-L1 on these cells [19]. Figure 3A,B show the high expression of both SIGLEC5 and PD-L1 on FHCs after a 5-day co-culture. Curiously, SIGLEC5 was expressed at high levels on CD14^+^ cells before co-culture (Figure 3A insert); however, neither CD14^+^ nor CD90^+^ cells exhibited PD-L1 on their cell surfaces before co-culture (Figure 3B insert). Next, the levels of the soluble form of these IC ligands were quantified in supernatants from sorted cells. Significantly higher levels of both sSIGLEC5 and sPD-L1 were found in cultures of hADMSCs and FHCs, suggesting these cells exhibited similar immunoregulatory properties (Figure 3C,D). In order to establish whether the increased expression of these immunomodulatory markers could aid in the resolution of inflammatory processes, the T cell lymphocyte proliferation was studied after a 5-day co-culture with sorted CD14^+^ monocytes, CD90^+^ hADMSCs, and CD90^+^CD14^+^ FHCs. FHCs exhibited a significantly greater ability to reduce both CD4- and CD8 T-cell proliferation compared to hADMSCs, showing their enhanced immunomodulatory properties (Figure 3E,F).

### 2.4. Foam Hybrid Cells Display High Migratory and Proliferative Abilities

To explore whether the fusion conferred any advantage to the FHCs in terms of migration, we performed a trans-well migration assay in response to FBS (Figure 4A). The percentage of migrated cells showed significant differences between FHCs and hADMSCs in active migration. FHCs migrated through the insert to a similar extent to human monocytes (Figure 4B). As previously observed in Figure 1C,D, confocal microscopy revealed that both hADMSCs and FHCs exhibited larger diameter sizes than monocytes; however, FHCs showed enhanced migratory abilities. Along these lines, the RT-qPCR analysis of the sorted populations illustrated that FHCs have a high expression of a matrix metalloproteinase (MMP) involved in MSC migration [20,21]. In contrast to MMP2, MMP9 was significantly upregulated in FHCs but not in hADMSCs (Figure 4C).

Additionally, to address the limited number of FHCs obtained, the proliferative capacities of these were studied. FHCs exhibited significantly greater proliferative abilities compared to their parental hADMSC (Figure 4D). To ensure the enhanced proliferation observed did not lead to the generation of tumours, sorted CD90^+^ hADMSCs and CD90^+^CD14^+^ FHCs were introduced to rats and left for 3 months. Appendix A shows the introduction of these cells did not generate tumours in vivo.

### 2.5. Foam Hybrid Cells Maintain Stem Cell Properties after Co-Culture

Finally, we moved on to study whether FHC maintained stemness properties. RT-qPCR analyses of sorted cells showed FHCs had significantly higher expressions of a number of pluripotency genes including *NANOG*, *c-Myc*, and *KLF4* compared to their parental (non-fused) cells (Figure 5A). As expected, hADMSCs showed a higher expression of these genes compared to human monocytes before co-culture (Figure 5A, inserts).

To test these stemness properties, after sorting, non-fused hADMSCs and FHCs were cultured for three weeks in two different conditioned media: adipogenic and chondrogenic, to test their differentiation abilities (Figure 5B). Brightfield microscopy pictures showed FHCs were able to differentiate into adipogenic (oil red positive) and chondrogenic cells (alcian blue positive), illustrating how FHCs maintained their parental stem cell properties of differentiation (Figure 5C,D). Additionally, the osteogenic differentiation abilities of sorted hADMSCs and FHCs were tested after 21 days following Alizarin red (AR) staining (Appendix A).

## 3. Discussion

The use of MSCs in regenerative medicine has shown promising preliminary results in human clinical trials [22]. Pre-clinical data demonstrating the immunomodulatory effects of MSC therapy in human and mouse in vitro culture generated optimism for the use of these cells as therapies in chronic inflammatory disorders [23,24]. However, the number of trials has steadily decreased since 2018 [25], as it has become an issue to reach phase III trials. Although phase I and phase II trials have shown some beneficial effects, the question still surrounds the fate of MSCs after administration [26,27].

There are several hypotheses regarding what occurs to these cells after infusion; few of them escape death, transfer their effects to another cellular entity, and exert their effects through release of trophic factors before death [9,28]. Nevertheless, none are capable of fully explaining the mechanism by which these MSCs exert their beneficial effects. Herein, we have found MSCs not only interact with the surrounding immune system, but also have the ability to fuse with human monocytes. These new hybrid entities, which we have denominated as FHCs, acquire enhanced mobility and proliferative abilities and keep both their parental immunomodulatory and stemness properties. These findings point towards a novel population which could aid in the understanding of what happens to MSCs after infusion.

Several authors have demonstrated the ability of MSCs to fuse with other cells, an event which has been widely studied in the cancer research field [29,30]. Studies have associated these hybrid MSCs with tumour progression and metastasis. Moreover, the immune system has also recently been associated with hybridisation events in the cancer field [18,31]. Although the hybridisation between MSCs and innate immune cells has not yet been postulated, studies have already pointed towards an interaction between the latter, which has been linked to an enhanced immunosuppressive phenotype of the MSCs [15]. MSCs are well known for supressing T cell proliferation, which is what makes them such interesting treatment options for sepsis, autoimmune diseases, and transplant medicine; however, the mechanisms of action for this phenomenon are poorly understood [32]. In this regard, a stronger reduction in T cell proliferation was observed after co-culture with FHCs, the latter of which we have also demonstrated to overexpress two IC ligands. These data propose FHCs to be potent immune modulators and novel treatment alternatives in MSC therapy.

The homing mechanism of MSCs is highly relevant to their migration towards injured sites. Herein, we demonstrated the enhanced migratory abilities of the resulting FHCs compared to their parental MSCs. These results have also already been described in the cancer field, where the hybridisation of cancer cells with human monocytes enhances migration abilities [18,31]. Furthermore, the migration of MSCs upon injury is mediated by the same soluble factors which recruit immune cells to the damaged area [33], once again pointing towards a strong interplay between MSCs and immune cells. Additionally, MSCs are known to have enhanced mobility through the overexpression of *MMP9*, which was found overexpressed in our FHCs [34]. Most importantly, these FHCs also maintained stemness properties, indicating their possible implication in regenerative medicine.

Eggenhofer et al. (2014) already highlighted the strong interplay which occurs between MSCs and immune cells under inflammatory conditions [9], data which is supported by our results, which show a greater generation of FHCs when MSCs are co-cultured with M1 polarised monocytes. However, these authors also questioned the rapid disappearance of MSCs after treatment and opened new paths towards explaining why these cells are no longer found after administration. Herein, our data propose hybridisation as a novel approach towards locating MSCs after infusion or even as an alternative population to administer after ex vivo generation.

Altogether, it is worth considering the fusion between monocytes and MSCs as an additional mechanism these cells have to carry their information to where it is most needed.

The fusion mechanism that may be behind metastasis [18,31] should be an aberrant behaviour of a physiological process. Although it is still highly speculative, perhaps our data indicate a physiological mechanism by which MSC cells manage to reach the sites where they are needed, acquiring mobility and proliferative capacity but without generating tumorigenic processes. Although additional experimentation is needed to demonstrate the role these FHCs may have, our data confirm the existence of a new hybrid entity which could fill the gap towards understanding the existence of heterologous MSC mechanisms of action in response to inflammation and after infusion.

## 4. Materials and Methods

### 4.1. Human Samples

Adipose tissue samples were obtained from healthy donors (n = 12) who underwent a liposuction process under the collaboration of the HULP plastic surgical service. These healthy donors met the following inclusion criteria: older than 18 years old and informed signed consent. Exclusion criteria: suffering from any relevant pathology and immunosuppressive treatment three months prior to surgery. This study was endorsed by the HULP Ethics Committee (Reference: PI1959).

### 4.2. Cell Culture

Human-adipose-derived mesenchymal stem cells (hADMSCs) were isolated from lipoaspirate or fat flaps following a standardised procedure [35].

Healthy donors were recruited from the Blood Donor Service of La Paz University Hospital. Peripheral blood mononuclear cells (PBMCs) were obtained by gradient centrifugation with Ficoll-Paque (GE Healthcare Bio-Sciences; Chicago, IL, United States). Monocytes were isolated from PBMCs after a one-hour adherence on Falcon^®^ 8-well culture slides (Corning; New York, NY, United States) on RPMI media (Gibco) supplemented with 0.01% penicillin/streptomycin (p/s) (Thermofisher; Waltham, MA, United States).

hADMSCs were co-cultured with human monocytes following a 1:10 ratio (hADMSC:Monocytes), following previous co-culture experimental conditions [18], on tissue-culture-treated Costar^®^ 24-well plates (Corning; New York, NY, United States) in RPMI (Gibco; Waltham, MA, United States), supplemented with 10% FBS (Gibco; Waltham, MA, United States) and 0.01% p/s (Thermofisher; Waltham, MA, United States) at 37 °C in a humidified atmosphere with 5% CO_2_ for 5 days. After co-culture time, cells were recovered by gentle scraping and analysed by flow cytometry or used for functional experiments.

### 4.3. Vital Colorant Assay

Monocytes were stained with DID vital colorant, and hADMSCs were stained with DIO vital colorant following the manufacturers’ instructions (Vybrant Multicolor Cell-Labelling Kit-Thermofisher). DID-stained monocytes were co-cultured with DIO-stained hADMSCs following a 1:10 ratio (hADMSCs:Monocytes) on Falcon^®^ 8-well culture slides (Corning; New York, NY, United States).

DID-Monocytes/DIO-hADMSCs co-cultures were fixed with 4% paraformaldehyde (PFA) (SIGMA-P6148) and mounted with Antifade Vectashield^®^ mounting medium (Vector) containing 4′,6-Diamidino-2-Phenylindole, Dihydrochloride (DAPI) for nuclear staining. Images were taken with a LeicaDMI400D confocal laser-scanning microscope at different magnifications. Excitation/emission wavelengths were 405/461, 644/665, and 484/501 nm for DAPI, DID, and DIO, respectively. All fluorescent image analyses were performed with LAS AF (Leica; Wetzlar, Germany).

### 4.4. Cell Sorting and Flow Cytometry Analysis

For cell sorting, co-cultured cells were stained in PBS (Gibco-BRL Life Technologies) with CD90-Allophycocyanin (APC) (Invitrogen; Waltham, MA, United States) and CD14-Fluorescein-isothiocyanate (FITC) (Immunostep; Salamanca, Spain) for 30 min in the dark at 4 °C at the manufacturers’ recommended concentration for each antibody. Afterwards, cells were washed with Phosphate Buffer Saline (PBS) and resuspended in RPMI supplemented with 10% FBS and 0.01% p/s media for cell sorting in BD FACS Influx TM Cell sorter (BD Biosciences; San Jose, CA, United States).

Fluorescence-Activated Cell-Sorting (FACS) analyses were developed using specific human antibodies (Abs) to the following surface molecules: CD90-APC (17-0909-42; 0.25 μg per test) (Invitrogen), PD-L1-Brilliant Violet (BV) 421 (563,738; 0.2 μg per test), HLA-DR-BV711 (563,696; 0.2 μg per test), CD163-Phycoerythrin (PE) (556,018; 1 μg per test) (all three from BD Biosciences), CD14-FITC (14F-100T; 1 μg per test) (Immunostep), and SIGLEC5-PE (5,190,523,078; 0.5 μg per test) (Miltenyi Biotec; Madrid, Spain).

Cells were stained with the indicated antibodies for 30 min at 4 °C in the dark and washed twice with PBS. Unstained samples were used as negative controls. For all assays, samples were run in a FACS Celesta (BD Biosciences) flow cytometer, and data were analysed using FlowJo (TreeStar; Chico, CA, United States) v.10.7.2 software.

### 4.5. Real Time Quantitative PCR Analysis

Sorted cells were washed twice with PBS, and the RNA was extracted using the High Pure RNA Isolation Kit (Roche Diagnostics; Basilea, Switzerland). The RNA concentration of each sample was measured using a NanoDrop 2000 (Thermo Fisher Scientific; Waltham, MA, United States). cDNA was synthesised from 0.25 μg total RNA using the High-Capacity cDNA Reverse Transcription Kit (Applied Biosystems; Waltham, MA, United States).

RT-qPCRs were performed using the QuantiMix Easy SYG Kit (Biotools; Madrid, Spain) according to the manufacturer’s instructions. Gene expression levels were analysed using the Biorad CFX96 touch deep-well real-time PCR detection system (Bio-Rad; Berkeley, CA, United States). Expression levels of *12S* housekeeping were used as an internal standard to normalise the data. Relative expression was determined using the 2(-Delta Delta C(T)) method. All primers were synthesised by Eurofins Genomics. Specific primers for each gene are shown in Appendix A.

### 4.6. In Vitro Modulation of Monocyte Polarisation Status

Isolated monocytes (Miltenyi Biotec Monocyte Isolation Kit II) were polarised towards the M1 phenotype with IFN-γ (100 µg/mL) and towards M2 with IL-4 (20 µg/mL) 48 h prior to co-culture with hADMSCs. Non-stimulated monocytes were used as control monocytes (M0).

### 4.7. Soluble Cytokines Quantification

Supernatants were collected after 48 h of monocyte polarisation. Cells were washed after 24 h and left resting for an additional 24 h. Cytokine levels (IL4, IL6, IL8, IL10, TNFα, and TGFβ) were measured in collected supernatants using the LegendPlex Human Essential Immune Response Panel, following the manufacturer’s instructions (Biolegend). Briefly, supernatants were incubated with premixed capture antibody-coated beads, washed, incubated with detection antibodies and streptavidin with phycoerythrin conjugate, acquired using a FACSCalibur flow cytometer (BD), and analysed with Biolegend v8.0 software (Biolegend; San Diego, CA, United States).

### 4.8. Soluble Immune-Checkopoint Quantification

Supernatants of sorted CD14^+^ monocytes, CD90^+^ hADMSCs, and CD90^+^CD14^+^ FHCs were collected after 5 days of cell culture. Soluble levels of PD-L1 were measured using the LegendPlex Human Immune Checkpoint Panel, following the manufacturer’s instructions (Biolegend; San Diego, CA, United States). Briefly, supernatants were incubated with premixed capture antibody-coated beads, washed, incubated with detection antibodies and streptavidin with s phycoerythrin conjugate (acquired using a FACSCalibur flow cytometer (BD)), and analysed with Biolegend v8.0 software (Biolegend; San Diego, CA, United States).

### 4.9. ELISA Assay

Soluble SIGLEC5 levels were determined using a commercially available enzyme-linked immunosorbent assay (ELISA) kit (Sigma-Aldrich; San Luis, CA, United States) in supernatants of sorted CD14^+^ monocytes, CD90^+^ hADMSCs, and CD90^+^CD14^+^ FHCs after 5 days. The manufacturer’s instructions were followed to determine these concentrations.

### 4.10. In Vitro Proliferation Assays

Human monocytes and hADMSCs were co-cultured for 5 days and taken to a sorter. After this, 5000 sorted CD14^+^ monocytes, CD90^+^ hADMSCs, and CD90^+^CD14^+^ FHCs were co-cultured with 2500 carboxyfluorescein succinimidyl ester (CFSE) labelled PBMCs from healthy volunteers (2:1 ratio, respectively). After 5 days, T-cell proliferation induced by pokeweed mitogen (2.5 µg/mL) was analysed by FACS. FACS analyses were developed using specific human antibodies (Abs) to the following surface molecules: CD4–PercP and CD8-APC (both from immunostep). Samples were run in a FACS Calibur (BD Biosciences) flow cytometer, and data were analysed using FlowJo (TreeStar; Chico, CA, United States) v.10.7.2 software.

### 4.11. Transwell Migration Assay

Co-cultures, monocytes, and hADMSCs were scraped after 5 days of culture, and cells were collected into RPMI (w/o FBS) and seeded (10^5^ cells) on the top compartment of Corning trans-well chambers (8 µm, Sigma; San Luis, CA, United States). The bottom compartment was filled with 700 µL RPMI + 10% FBS. After 48 h, the contents from the top and bottom were recovered and analysed through flow cytometry using 1000 inert beads (Reagent D from Mice CBA, BD Biosciences) as a control to stop acquisition. The migration (M) percentage was assessed as the count of each cell type in the bottom (B) with respect to the remaining amount in the top (A), as shown in: M = [B/(A + B)] × 100.

### 4.12. hADMSC Differentiation and Staining

For hADMSC differentiation, cells were cultured in 6 multi-well plates with MEMalpha 10% FBS and 0.1% p/s until reaching 70% of confluence. Once reached, cells were treated for 21 days in order to induce differentiation. Briefly, the culture media were supplemented with 1 µM dexamethasone, 500 µM IBMX (3-isobutyl-1methylxanthine, from Sigma-Aldrich), 100 µM indomethacin (Sigma-Aldrich), and 10 µg/mL insulin (Novo Nordisk; Bagsvaer, Denmark) for adipogenic induction or 50 µM L-ascorbic acid 2-phosphate (Sigma-Aldrich; San Luis, CA, United States), 6.25 µg/mL insulin and 10 ng/mL TGFβ-1 (Peprotech; Waltham, MA, United States) for chondrogenic induction or 10% calf serum (Thermo-Fisher Scientific), 2.5 mg/L of Amphotericin, 10 mM of beta glycerophosphate (both from Merck), 100mg/mL Penicillin/Streptomycin (Gibco; Waltham, MA, United States), 0.1 µM dexamthasone, 2.5 mg/L ofascorbic acid-2-phosphatase (both from Sigma-Aldrich). The medium was renewed every 3 days. After 21 days of differentiation, Red Oli O (Sigma-Aldrich), Alcian Blue 8 GX, and Alizarin Red S (both from Sigma-Aldrich) specific stainings were performed to analyse adipogenesis, chondrogenesis, and osteogenesis, respectively. Briefly, for adipogenesis, cells were fixed with 10% formaldehyde 10 min at room temperature (RT), then washed with PBS, and stained with filtered fresh solution of 3:2 (*v*/*v*) ddH_2_O and 0.5% Oli red O in isopropanol for 15 min. For chondrogenesis, cells were fixed with methanol (2 min, 20 °C), rinsed once with ddH_2_O, and then stained overnight at RT with 0.1% Alcian blue in 0.1 N HCl. Finally, for osteogenesis, cells were carefully washed with PBS and fixed with 10% formaldehyde for 30 min, then cells were washed with distilled water, and Alizarin Red S staining solution (2 g dissolved in 100 mL of distilled water) was added to cover the cell monolayer for 45 min. To finalise, Alizarin Red S solution was washed with distilled water, and PBS was added to the cell monolayer. Photos were taken at 20× magnification.

### 4.13. Animals

The experiments were conducted using 6 healthy male Sprague-Dawley rats. All rats weighed between 200 g and 250 g at the start of the experiments and were 9–10 weeks old. The rats were kept in a climate-controlled environment where the temperature was kept constant at 20 °C ± 3 °C. The rats had ad libitum access to standard rat chow and water. For the cell therapy administration, 5000–8000 sorted cells (CD90^+^CD14^+^ and CD90^+^, respectively) were resuspended in 1 mL of saline. The animals were anaesthetised using sevoflurane (Sevorane^®^, Abbott Laboratories; Chicago, IL, United States) at 3%. Cell suspensions were injected in the tail vein. The animals were sacrificed 3 months after cells administration. For this purpose, the animals were anaesthetised using sevoflurane at 6%, and the heart was stopped by injecting KCL directly into it. The death of the animal was confirmed by cutting the spinal cord. Finally, an incision in the intraperitoneal cavity was completed, and the required organs were obtained.

### 4.14. Immunohistochemistry

Tissues were fixed in 4% paraformaldehyde (SIGMA-P6148), embedded in paraffin and cut into 5 μm sections with a Leica RM2255 microtome (Leica Biosystems; Wetzlar, Germany). Haematoxylin and Eosin (both from Merck) staining was performed following the manufacturer’s instructions (Merck; Darmstadt, Germany). Pictures of stained sections were taken at 10× magnification on an Olympus Bx41 microscope.

### 4.15. Statistical Analysis

Continuous variables were analysed by either parametric or non-parametric statistical tests after analysing their Gaussian distributions using the Saphiro–Wilk test. Accordingly, t-student and ANOVA tests followed by Tukey analysis or Kruskall–Wallis tests were used. Significance was set (* *p* < 0.05, ** *p <* 0.01, *** *p* < 0.001, **** *p* < 0.0001) using Graphpad Prism 9.0 software.

### 4.16. Ethics Approval

All volunteers signed informed consents, and data were treated according to recommended criteria of confidentiality, following the ethical guidelines of the 1975 Declaration of Helsinki. The study was approved by the HULP Ethical Committee (Reference: PI1959).

## Figures and Tables

**Figure 1 ijms-23-09672-f001:**
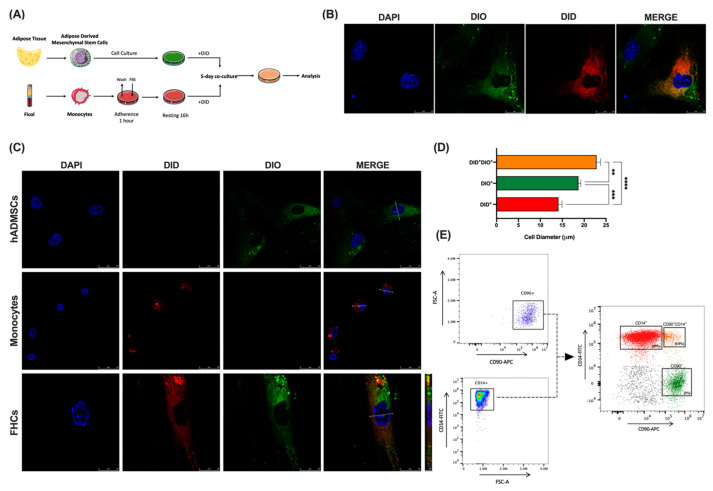
Human-adipose-derived mesenchymal stem cells fuse with human monocytes ex vivo. (**A**), Schematic representation of experimental co-culture conditions. (**B**), Representative confocal images at 63× magnification of co-cultured DIO stained hADMSCs (green) with DID stained monocytes (red) and DAPI for nuclei (blue) is shown. Merges show co-cultured cells DID^+^DIO^+^ (tangerine). (**C**), Representative confocal images at 63× magnification of DIO stained hADMSCs (green), DID stained monocytes (red) and hybrid cells (FHCs) (tangerine) are shown. DAPI was used for nuclei labelling (blue). Diameter sizes are shown in merge panels. (**D**)**,** Diameter comparisons between hADMSCs (green), monocytes (red), and FHCs (tangerine) are shown (n = 3). (**E**), Left panels, representative FACS analysis gating strategy illustrating CD90^+^ and CD14^+^ single staining’s. Right panel, dot plot after 5-day co-culture illustrates CD90^+^ hADMSCs (green), CD14^+^ human monocytes (red), and CD90^+^CD14^+^ (FHCs, tangerine). **, *p <* 0.01; ***, *p <* 0.001; ****, *p <* 0.0001 in one-way ANOVA test with Tukey’s multiple comparison post-hoc test.

**Figure 2 ijms-23-09672-f002:**
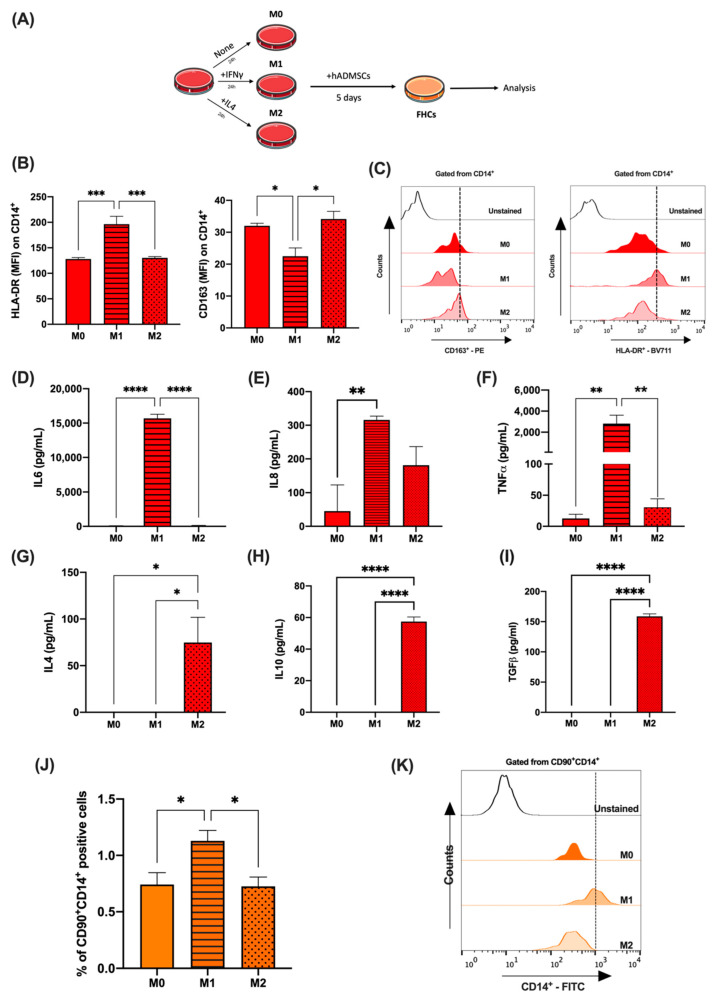
Human monocyte polarisation to M1 yields a greater number of FHCs. (**A**), Schematic representation of experimental monocyte polarisation and co-culture conditions. (**B**), Mean fluorescence intensity (MFI) of surface markers CD163 and HLA-DR on control, M1 (dashed bar) and M2 (dotted bar) polarised monocytes (n = 4). (**C**), Representative FACS histograms of CD163 and HLA-DR fluorescence intensity gated from CD14^+^ Unstained (clear) M0 (dark red) M1 (red) and M2 (light red) human monocytes are shown. Cytokine levels of (**D**), IL6; (**E**), IL8; (**F**), TNFα; (**G**), IL4; (**H**), IL10 and (**I**), TGFβ in cell supernatants from control, M1 (dashed bar) and M2 (dotted bar) polarised monocytes (n = 4). (**J**), Percentages of CD90^+^CD14^+^ cells of hADMSCs cocultures with M0, M1 (dashed bar) and M2 (dotted bar) polarised monocytes (n = 4). (**K**), Representative FACS histograms of CD14 expression in gated CD90^+^C14^+^ cells from hADMSCs cocultures with M0, M1 and M2 monocytes. *, *p <* 0.05; **, *p <* 0.01; ***, *p <* 0.001; ****, *p <* 0.0001 in one-way ANOVA with Tukey’s multiple comparison post-hoc test.

**Figure 3 ijms-23-09672-f003:**
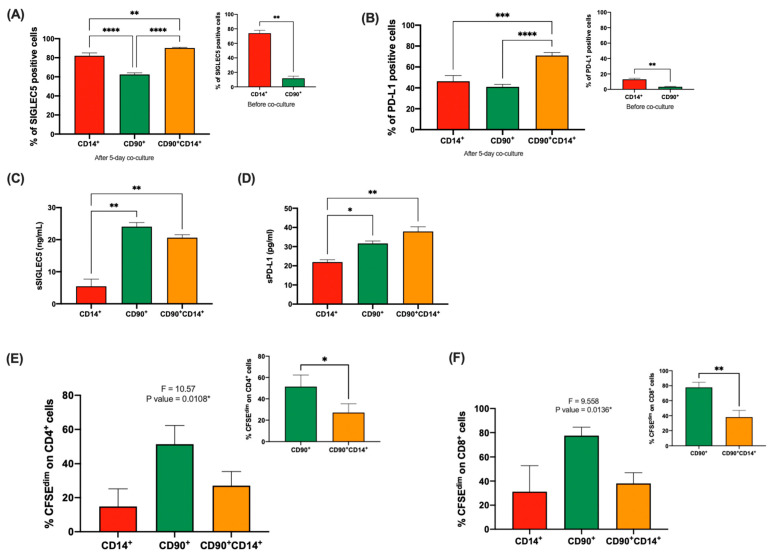
Immunomodulatory properties of novel-resulted FHCs after co-culture. (**A**), Analysis of SIGLEC5 expression on CD14^+^ human monocytes (red), CD90^+^ hADMSCs (green), and CD90^+^CD14^+^ FHCs (tangerine) after 5-day co-culture by FACS (n = 9). (Insert), Analysis of SIGLEC5 expression on CD14^+^ human monocytes (red) and CD90^+^ hADMSCs (green) before co-culture by FACS. **, *p <* 0.01; ***, *p <* 0.001; ****, *p <* 0.0001 in one-way ANOVA with Tukey’s multiple comparison post-hoc test and **, *p <* 0.01 in Mann–Whitney test, respectively. (**B**), Analysis of PD-L1 expression on CD14^+^ human monocytes (red), CD90^+^ hADMSCs (green), and CD90^+^CD14^+^ FHCs (tangerine) after 5-day co-culture by FACS. (Insert), Analysis of PD-L1 expression on CD14^+^ human monocytes (red) and CD90^+^ hADMSCs (green) before co-culture by FACS. **, *p* < 0.01; ***, *p <* 0.001; ****, *p <* 0.0001 in ordinary one-way ANOVA with Tukey’s multiple comparison post-hoc test and **, *p <* 0.01 in Unpaired *t*-test, respectively. Quantification of soluble immune-checkpoint ligands levels of (**C**), sSIGLEC5 and (**D**), sPD-L1 in cell supernatants from sorted CD14^+^ human monocytes (red), CD90^+^ hADMSCs (green), and CD90^+^CD14^+^ FHCs (tangerine) cultured along for 120 h. *, *p <* 0.05; **, *p <* 0.01 in ordinary one-way ANOVA with Tukey’s multiple comparison post-hoc test. Proliferation of (**E**), CD4 and (**F**), CD8 T lymphocytes after a 5-day co-culture with pokeweed (PWD) stimulated sorted monocytes (CD14^+^, red), hADMSCs (CD90^+^, green) and FHCs (CD90^+^CD14^+^, tangerine). *, *p* < 0.05 in one-way ANOVA with Tukey’s multiple comparison post-hoc test. Inserts show T lymphocyte proliferation comparison between hADMSCs (CD90^+^, green) and FHCs (CD90^+^CD14^+^, tangerine) for CD4^+^ (left) and CD8^+^ (right). *, *p* < 0.05 and **, *p <* 0.01 in unpaired *t*-test.

**Figure 4 ijms-23-09672-f004:**
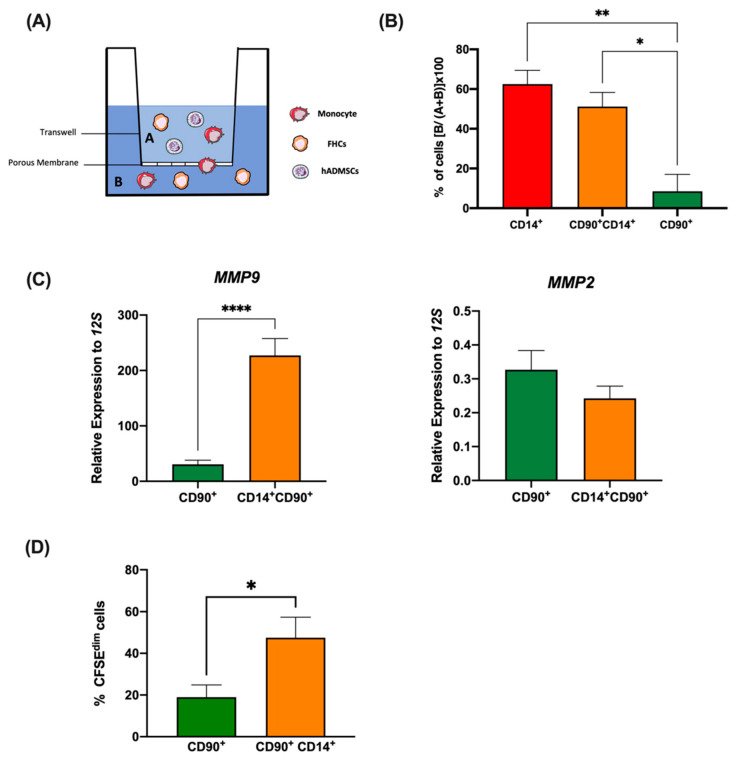
FHCs acquire enhanced migratory and proliferative abilities. (**A**), Schematic representation of the migration assay. Cells resulted from fusion assay were put on the top chamber (A), then migrating cells were quantified in the bottom chamber (B) by FACS. (**B**), Percentages of migration, % of cells [B/(A + B)] × 100 of CD14^+^ human monocytes (red), CD90^+^ hADMSCs (green), and CD90^+^CD14^+^ FHCs (tangerine) from the trans-well assay are shown (n = 4). *, *p* < 0.05; and **, *p* < 0.01 in ordinary one-way ANOVA with Tukey’s multiple comparison post-hoc test. (**C**), Relative expression by RTqPCR of *MMP9* and *MMP2* in CD90^+^ hADMSCs (green) and CD90^+^CD14^+^ FHCs (tangerine) sorted cells after 5 days of co-culture (n = 9). ****, *p <* 0.0001 in unpaired *t*-test. (**D**), Proliferations of CD90^+^ hADMSCs (green) and CD90^+^CD14^+^ FHCs (tangerine) are shown. *, *p* < 0.05 in unpaired *t* test.

**Figure 5 ijms-23-09672-f005:**
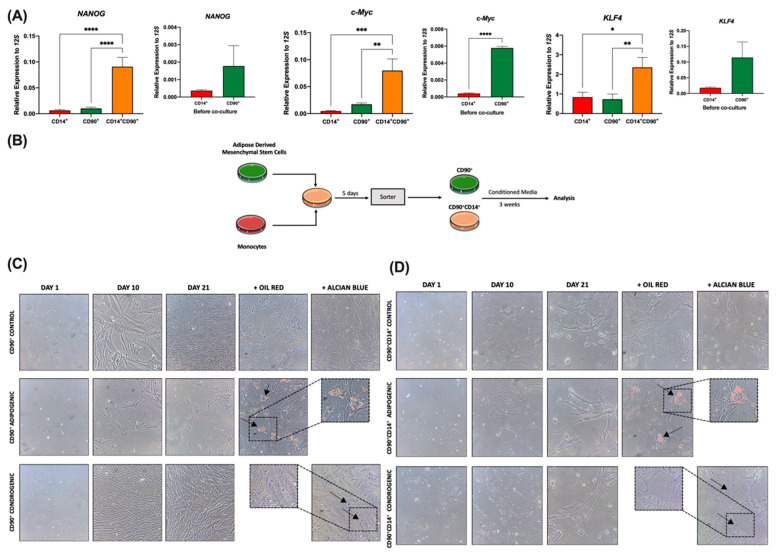
FHCs maintain stemness properties after fusion. (**A**), Main panels, the relative expression of *NANOG*, *c-Myc*, and *KLF4* genes by RTqPCR of sorted CD14^+^ human monocytes (red), CD90^+^ hADMSCs (green), and CD90^+^CD14^+^ FHCs (tangerine) after 5 days of co-culture are shown (n = 9). *, *p* < 0.05; **, *p* < 0.01; ***, *p* < 0.001 and ****, *p* < 0.0001 in ordinary one-way ANOVA with Tukey’s multiple comparison post-hoc test. (Inserts), the relative expression of *NANOG, c-Myc* and *KLF4* genes of CD14^+^ human monocytes (red) and CD90^+^ hADMSCs (green) before co-culture are shown (n = 3). ****, *p* < 0.0001 in unpaired *t* test. (**B**), Schematic representation of experimental conditions for cellular differentiation. (**C**), Representative brightfield microscopy images for kinetics of cellular differentiation of CD90^+^ hADMSCs images at 1, 10, and 21 days at 20× magnification. Oil red was used to confirm adipogenic differentiation and alcian blue to determine chondrogenic differentiation. Arrows indicate differentiation areas. (**D**), Representative brightfield microscopy images for kinetics of cellular differentiation of sorted CD90^+^CD14^+^ cells (FHCs) images at 1, 10, and 21 days at 20× magnification. Oil red was used to confirm adipogenic differentiation and alcian blue to determine chondrogenic differentiation (n = 3). Arrows indicate differentiation areas.

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
