# Peer review of "Fused Cells between Human-Adipose-Derived Mesenchymal Stem Cells and Monocytes Keep Stemness Properties and Acquire High Mobility"

_ijms, 2022, doi:10.3390/ijms23179672_

Round 1
Reviewer 1 Report
The researchers in this study described cell fusion between hADMSCs and monocytes ex vivo forming Foam Hybrid Cells (FHCs). These cells exhibited strong immunomodulatory features, displayed high migratory and proliferative abilities, and maintained stem cell properties This new hybrid cell population could provide a novel explanation for the lack of MSCs detection after administration.
The work is well conducted with appropriate controls and I recommend the paper could be published in the “IJMS” after the authors address the specific points mentioned below.
Addressing the comments will improve the quality of the manuscript and the impact of this research work.
Specific points:
1. Line 4: Please delete ‘fullstop’ at the end of the title.
2. Line 67 and 76: “studies” instead of “authors”.
3. Line 256: Why OCT4 and SOX2 expression (which are Yamanaka factors) was not investigated alongwith NANOG, c-Myc and KLF4? The authors should show expression of these two Yamanaka factors as well. The authors should delete “Yamanaka’s” and instead write “pluripotency genes” as Nanog is not a Yamanaka factor but a Thomson factor.
4. Line 262: Why only adipogenic and chondrogenic differentiation of FHCs was investigated? Why the authors did not investigate osteogenic differentiation by Alizarin Red staining? Did they see any difference? The authors should also perform this experiment.
5. Also, the stainings of adipogenic and chondrogenic differentiation are not good as shown in other manuscripts.
6. Line 268: Should be “Figure 5” instead of “Figure 4”.
7. Line 303: “transplant medicine” instead of “trasplant medicine”.
8. Line 370-373: Please provide catalog numbers and concentrations of antibodies used in the study.
Author Response
- Line 4: Please delete ‘fullstop’ at the end of the title.
The full stop at the end of the title has been removed.
- Line 67 and 76: “studies” instead of “authors”.
This change has been made in the revised version of the manuscript.
- Line 256: Why OCT4 and SOX2 expression (which are Yamanaka factors) was not investigated along with NANOG, c-Myc and KLF4? The authors should show expression of these two Yamanaka factors as well. The authors should delete “Yamanaka’s” and instead write “pluripotency genes” as Nanog is not a Yamanaka factor but a Thomson factor.
We thank the reviewer for their appreciation. No significant differences were observed regarding OCT3/4 and SOX2, therefore this data was not included. Considering this, Yamanakas cassette has been replaced for pluripotency genes in the revised version of the manuscript.
- Line 262: Why only adipogenic and chondrogenic differentiation of FHCs was investigated? Why the authors did not investigate osteogenic differentiationby Alizarin Red staining? Did they see any difference? The authors should also perform this experiment.
Osteogenic differentiation for sorted hADMSCs and FHCs was also tested with the use of Alizarin Red staining. No differences were found either macroscopically or microscopically between sorted hADMSCs and FHCs. These results had not been previously included in the manuscript because this experiment was performed separately due to the number of cells obtained after sorting and was only evaluated at day 21, thus not allowing the results to be included with those in Figure 5. Nonetheless, these results have been included in Supplementary Figure 3 in the revised version of the manuscript. The methods followed for this osteogenic differentiation have also been included in the materials and methods section of the revised manuscript.
- Also, the stainings of adipogenic and chondrogenic differentiation are not good as shown in other manuscripts.
We thank the reviewer for their comment. Although there is a zoomed image of this in the manuscript, we have included in this rebuttal images of these staining of greater size to make it easier to appreciate the differentiations. In the case of Figure 5, these images are smaller in order to allow all controls to fit into a single Figure. It is also important to note these are sorted cells and therefore there is presence of artefacts in the medium which also influence and worsen the quality of these images compared to other manuscripts.
(please find the figure in the document attached)
- Line 268: Should be “Figure 5” instead of “Figure 4”.
This mistake has been corrected in the revised version of the manuscript.
- Line 303: “transplant medicine” instead of “trasplant medicine”.
This mistake has been corrected in the revised version of the manuscript.
- Line 370-373: Please provide catalog numbers and concentrations of antibodies used in the study.
The catalog numbers and concentrations of these antibodies have been included in the revised manuscript.
All the changes described in each point have been highlighted for easy follow-up.

Reviewer 2 Report
Thank you for giving me this opportunity to review the research article entitled, "Fused cells between human adipose derived mesenchymal stem cells and monocytes keep stemness properties and acquire high mobility".
I here carefully reviewed the submitted set of the manuscript and found it merits of publication.
However, the Discussion section seems very unclear and immature, just introducing the results repetitively obtained from the present study. I can't evaluate the novel findings as the existence of a new hybrid entity with the fusion between monocytes and MSCs, which could fill the gap towards understanding MSC fate after treatment, without discussing the findings from the present study referring to the up-to-date evidences from the literature.
Author Response
I here carefully reviewed the submitted set of the manuscript and found it merits of publication. However, the Discussion section seems very unclear and immature, just introducing the results repetitively obtained from the present study. I can't evaluate the novel findings as the existence of a new hybrid entity with the fusion between monocytes and MSCs, which could fill the gap towards understanding MSC fate after treatment, without discussing the findings from the present study referring to the up-to-date evidences from the literature.
We thank the reviewer for their comment. We have worked around the previous discussion and have included up-to-date evidences which support our findings and allow for a better discussion of the results obtained. We believe with the changes made in the revised version you will find the discussion much clearer. All changes have been highlighted for easy follow-up.

Round 2
Reviewer 1 Report
The manuscript can now be accepted since the authors have addressed the comments of the reviewers.
Reviewer 2 Report
Thank you very much for giving me this opportunity to review this study article.
I here carefully re-reviewed the revised set of the manuscript and found it merits for publication in the present form.
Congratulation on the authors.